# Recent Studies on Fluorinated Silica Nanometer-Sized Particles

**DOI:** 10.3390/nano9050684

**Published:** 2019-05-02

**Authors:** Scott T. Iacono, Abby R. Jennings

**Affiliations:** Department of Chemistry and Chemistry Research Center, United States Air Force Academy, 2355 Fairchild Dr, Colorado Springs, CO 80840, USA; scott.iacono@usafa.edu

**Keywords:** fluorinated silicas, silica nanoparticles, low-surface energy, sol-gel method

## Abstract

Since initially being reported, fluorinated silica nanometer-sized particles (F-SiNPs) have gained much interest in the scientific community, due to their unique properties. These properties, include, low surface energies, increased mechanical strength, thermal robustness, and chemical resistance, and are a direct result of the incorporation of fluorine with a nanometer-sized silica network. This review aims to summarize the synthetic methods that have, and are still, being utilized to prepare these specialized materials. Following this, applications for F-SiNPs, with an emphasis on recent examples, will be presented in further detail.

## 1. Introduction

Fluorinated materials and polymers have found use in a number of advanced applications, including imaging [1,2], electronics/optoelectronics [3,4,5], conductors [6,7,8], pharmaceuticals [9,10], biomedical [11], surface coatings [12,13,14], automotive, and the aerospace industry [15]. This can be directly attributed to the unique properties fluorine atoms impart on the resulting functional materials. Fluorine is a well-known electronegative atom, with a value of 4.00 on the Pauling scale, making the C–F bond highly polarizable and quite stable [16]. Additionally, fluorine is considered a poor leaving group in most cases [17]. As a result of these factors, fluorinated materials generally exhibit unprecedented thermal and chemical stabilities. Although significantly weaker than hydrogen bonds formed between well-recognized hydrogen-bond acceptors (N, O, F), it is known that organofluorine compounds (containing C–F bonds) can interact with various aliphatic compounds (containing H–C bonds) in a similar fashion, as a very weak hydrogen-bond acceptor [18]. This plays an important role in the structural arrangement and properties of these materials. Fluorine also has a high ionization energy and one of the lowest polarizability values, giving fluorinated materials increased hydrophobicity, lipophilicity, and low refractive indexes [19]. Despite these attractive qualities, fluorinated materials and polymers are often plagued by inferior mechanical properties and poor adhesion, limiting the scope of their applications [20,21].

To combat this, it is common practice to formulate hybrid composite materials, which combine the attractive properties of fluorinated materials, with a reinforcing or stabilizing material that offers mechanical robustness and better adhesion. Reinforcing materials that have been utilized in the past with fluorinated materials include ceramics [22], carbon nanomaterials [23], polymers [24], and various types of silicas [25]. These agents can be incorporated into the fluorinated material or polymer as a physically blended composite or through covalent attachment via reactive end groups. Either way, the filler component must be compatible and dispersible within the fluorinated component, in order to capitalize on the desired properties, while limiting or preventing aggregation. 

When identifying an appropriate reinforcing agent, silica-based materials offer some benefits over the other types of materials. To begin, there are a wide range of commercially available products, which include silsesquioxanes and various sized particles, that can be used as received. Additionally, silica-based materials can be readily synthesized in-house, using slightly modified solution-gelation (sol-gel) conditions from inexpensive, commercially-available materials. This is a very straightforward synthetic route when compared to those used for other stabilizing agents, which generally require harsh reaction conditions, tedious purifications, and complex processing techniques [26,27,28]. Furthermore, the sol-gel method offers a simple platform for introducing additional functionalities, which is key for tailoring material properties. Silica-based materials are also chemically stable, thermally robust, and compatible with a variety of fluorinated-organic based materials and polymers [29]. They can also be physically blended or covalently attached to a fluorinated component in order to obtain specialty composites. Within the arena of silica-based reinforcing agents, silica nanometer-sized particles (SiNPs) are ideal candidates, as not only do they exhibit the aforementioned attributes, but there is also a profound effect on the properties of the particles, due to their nanometer size range. With a large surface area-to-volume ratio, and as the size of the particles decrease, more atoms reside at the surface, and surface properties tend to dominate. These properties can then be imparted to the material they are incorporated into. Furthermore, when nanometer sized fillers are utilized, they are known to yield materials and composites with micro- and nanotextured surfaces. Although explained in more detail in the subsequent sections, the production of nanotextured surfaces give rise to enhanced surface roughness and are an essential property for many of the specialized applications for F-SiNPs. Silica nanometer-sized particles have also been found to disperse better than other reinforcing agents [30]. The particles can be purchased or synthesized with discrete and uniform dimensions, which is important for maintaining consistency throughout a functional material. Furthermore, with the ease in which covalent attachment can be achieved via sol-gel techniques, the ability to obtain F-SiNPs is virtually unmatched.

In the subsequent paragraphs, the field of F-SiNPs will be described, with emphasis on the synthetic methods utilized to obtain the materials and selected applications. Due to the vast number of scientific papers on this type of fluorinated hybrid material, this article is not meant to be a complete comprehensive review, but a summary of recent examples and how they contribute to an already well-established field.

## 2. Synthetic Methods 

There are a number of different synthetic strategies utilized for preparing F-SiNPs, with the most heavily utilized being sol-gel methods. The goal of this section is to provide the reader with an overview of the sol-gel technique, with selected recent examples, while emphasizing the lesser employed processes. 

### 2.1. Sol-gel Methods

The foundation for employing sol-gel chemistry to obtain F-SiNPs is rooted in the knowledge that reactive silanes, including alkoxysilanes, halosilanes, and silanols will easily undergo hydrolysis and condensation [31]. Hydrolysis can be achieved in a number of ways, including heat, excess water, and acid or base catalysis. Sol-gel techniques can be used in a “top-down” or “bottom-up” fashion. Scheme 1 shows a general top-down and bottom-up method for the preparation of F-SiNPs from alkoxysilanes. 

In a top-down method, the reactive silane, containing the fluorinated component (R_f_, Scheme 1), or a reactive end group that the fluorinated component can be attached to or synthesized from, will be hydrolyzed. Hydrolysis results in the formation of silanol groups that will readily undergo condensation with other reactive silanes and the silicon nanoparticles, which are also known to contain surface silanols. If the fluorinated component will be incorporated through a reactive end-group, it is essential to ensure that the end group is latently reactive, i.e., it remains in tack and is unaffected by the sol-gel conditions employed in the functionalization step. Complimentary to this approach is the bottom-up technique. In this method, the fluorinated component is contained in a reactive silane and is typically co-hydrolyzed/condensed with a different reactive silane. The example shown in Scheme 1 is the co-hydrolysis/condensation of a fluorinated (R_f_) triethoxysilane, with tetraethyl orthosilicate (TEOS). It should be noted that other reactive silanes, besides TEOS and a functionalized triethoxysilane, can be employed. By tailoring the reaction conditions, F-SiNPs can be obtained. 

#### Selected Examples

Recently, our research group prepared a reactive silane monomer, containing perfluorocyclopentene (PFCP) [32]. The fluorinated reactive silane reagent prepared was unique in that, with the PFCP ring containing a vinylic fluorine, it not only serves as a means for the incorporation of fluorine, but also a latent reactive end group. Thus, we were able to use the monomer to functionalize SiNPs (prepared in house) that were 342 nm in diameter, by a top-down method. Since the vinylic fluorine atom within the PFCP group remains intact during the functionalization, it could be utilized to post-synthetically modify the particles with hexafluoroisopropanol (HFIPA, HOCH(CF_3_)_2_) in order to incorporate more fluorine and change the surface energy of the functionalized particles, Scheme 2. Water contact angle (WCA) measurements were utilized to confirm the functionalization and post-synthetic modifications of the particles, and were found to be 139°, and 135°, respectively. 

Chakrabarty and co-workers also synthesized F-SiNPs by employing a top-down method [33]. Initially, a water soluble macro- reversible addition-fragmentation chain-transfer (RAFT) agent, containing triethoxysilane, was prepared and then utilized to make a fluorinated polymer in a surfactant-free emulsion by means of polymerization-induced self-assembly. Figure 1 shows the macro-RAFT agent (P(4VP salt-*co*-VTES)) and fluorinated polymer ((P(TFEMA)) that was prepared. 

Once synthesized, the fluorinated polymer containing the reactive triethoxysilane groups was used to functionalize commercially acquired SiNPs with a diameter of 22 nm, via acid catalyzed hydrolysis and condensation. The functionalization of the SiNPs was confirmed by Fourier-transform infrared (FT-IR) spectroscopy and transmission electron microscopy (TEM). The authors then utilized the F-SiNPs to prepare superhydrophobic films. The films had WCA of 151.5°, Scheme 3. 

In another example, Mori et al. implemented a bottom-up technique by co-hydrolyzing/condensing (3-mercaptopropyl)triethoxysilane and two different fluorine containing reactive silanes, Figure 2 [34]. An analysis of the particles by ^19^F nuclear magnetic resonance (NMR) spectroscopy showed successful incorporation of the fluorinated component and scanning force microscopy indicated the structures where uniform and between 4–6 nm. With the F-SiNPs also containing thiol end groups, the authors prepared UV-cured hybrid films that showed promise for UV-induced nanoimprinting. 

### 2.2. Additional Synthetic Methods

As already mentioned, the most commonly reported synthetic method utilized to obtain F-SiNPs is by far the versatile sol-gel technique. Although rarely employed, there are a few additional techniques worth mentioning. For example, Marichal and co-workers were able to fluorinate SiNPs, with a diameter of 16 nm, by stirring them for 24 h in aqueous ammonium fluoride (NH_4_F) [35]. The stirred solutions had a molar composition of SiO_2_:NH_4_F:H_2_O of 1:0.015–0.6:16.8, respectively. Using ^19^F, ^29^Si, and ^1^H magnetic angle spinning (MAS) NMR, along with ^1^H-^29^Si and ^19^F-^29^Si cross-polarization MAS NMR spectroscopy, the authors determined two different fluorination mechanisms were likely occurring and were dependent on NH_4_F concentration. At low concentrations, a nucleophilic substitution of the hydroxyl groups by the fluoride ion occurs, while at high concentrations, a second mechanism also occurs with the fluoride ion opening siloxane bonds, Scheme 4. According to the authors, this was further supported by the four types of fluorosilicate species that were detected throughout the analysis. 

More recently, Ilharco and Lopes prepared F-SiNPs using a reverse emulsion method; a mixture of cyclohexane, Triton X-100, hexanol, and aqueous hydrofluoric acid (HF), with the sol-gel reactions of TEOS and (3-aminopropyl)triethoxysilane (APTES) [36]. This yielded hollow F-SiNPs, with an external diameter of 37 nm. An analysis of the particles by diffuse reflectance infrared Fourier-transform (DRIFT) spectroscopy showed successful fluorination of the particles by aqueous HF. In addition, the authors used similar procedures to post-synthetically fluorinate bulk SiNPs. An analysis of the particles by DRIFT spectroscopy showed similar results to the hollow spheres, that aqueous HF could be used to fluorinate the SiNPs, with a strong Si-F band at 741 cm^−1^, Figure 3a. Transmission electron microscopy indicated partial destruction of the particles as a result of the HF treatment, resulting in the roughening of the surfaces, Figure 3b. 

## 3. Selected Applications

As discussed in the Introduction section, F-SiNPs are hybrid materials with unique characteristics and properties. As a result, these specialty composites have found use in a variety of applications. This section is meant to summarize some of those applications, with an emphasis on more recent examples. 

### 3.1. Advanced Surface Coatings 

The most heavily reported application for the utilization of F-SiNPs is by far for advanced surface coatings. These types of hybrid materials are often employed for self-cleaning, non-fouling, abrasion resistant, or stain-resistant coatings as a result of their unique and distinctive wetting properties. In order to determine the wettability of a surface, contact angle measurements and contact angle hysteresis, with a variety of different liquids, are often investigated [37]. The wetting of these specialized coatings can range from hydrophobic to hydrophilic, oleophobic to oleophilic, or omniphobic to omniphilic [38,39]. Furthermore, a surface can display extreme or super-repellency towards a liquid, for example superhydrophobicity, if the contact angle is greater than 150° and the hysteresis is lower than 5° [40]. Factors affecting the wettability include the surface energy and the structural morphology of the surface [41,42]. The surface energy can be tailored by utilizing the chemistry or functional groups present on the surface of the coating. In the case of F-SiNPs, the incorporation of the fluorine atoms results in the generation of materials with very low surface energies. Additionally, fluorine has been shown to migrate towards the surface [43]. Employing F-SiNPs in the surface coating also results in micro- and nanotextured surfaces with an increased surface roughness [44]. As a result of these factors, the production of super-repellent surfaces can be achieved [38,45]. 

In a recent example presented by He et al., the authors utilized F-SiNPs to prepare hydrophobic adhesive coatings [46]. In a top-down method, F-SiNPs were prepared by first functionalizing bare silica nanoparticles with APTES. Next, fluorinated glycidyl copolymers were grafted to the particles after attachment of an atom transfer radical polymerization (ATRP) initiator, Figure 4a. The authors investigated three different formulations of the F-SiNPs, in which the amount (mols) of the glycidyl block was varied. A control was also prepared that contained no fluorine. As seen in Figure 4b, the functionalized particles had an average diameter of 120 +/− 10 nm. The F-SiNPs were cast onto glass to investigate the hydrophobicity, surface energy, and adhesive strength of the coatings. The authors determined the surface coatings prepared from the F-SiNPs had increased hydrophobicity by showing higher WCA (114–119° vs. 90° for the control) and more resistance to water absorption. This was directly attributed to the surface migration of the fluorine, as well as, the increased surface roughness from the SiNPs.

Furthermore, the adhesive strength of the agglutinated film, with the highest fluorine content (by wt %) and surface roughness, was evaluated under humid thermal aging conditions with respect to the control, Figure 4c. Although the control had better adhesive strength initially than the F-SiNPs film (1.92 MPa vs. 1.82 MPa), under the humid thermal aging conditions, the F-SiNPs film was found to maintain its adhesive properties after 30 cycles; whereas the control had decreased by nearly 20%. This was directly attributed to the aging resistance of the fluorine in the F-SiNPs.

In another example presented by Lin et al., the authors prepared anti-oil-fouling superoleophobic (SHI)–superhydrophilic (SOP) cotton fabrics utilizing F-SiNPs [47]. The coatings were prepared by dip-coating the cotton fabric in two different solutions; the first containing bare SiNPs with a diameter of 200 nm, and the second containing a reactive fluorinated phosphate oligomer and 1*H*,1*H*,2*H*,2*H*-perfluorodecyltriethoxysilane (PFTES). In addition to containing a fluorinated block, the phosphate oligomer contained a polyethylene glycol (PEG) block, giving it both hydrophobic and hydrophilic properties. An analysis of the surface chemistry of the fabrics pre- and post-dip-coating by FT-IR and X-ray photoelectron spectroscopy (XPS) indicated that the cotton fabric had been successfully functionalized. Furthermore, scanning electron microscopy (SEM) and atomic force microscopy (AFM) analysis showed a clear change in the morphology of the cotton fibers and an increase in the root mean square (RMS) surface roughness, as a result of the dip coating, Figure 5a. The authors then evaluated the cotton fabrics wetting properties with a variety of oils and water by contact angle analysis. The coated cotton fabrics were determined to be SHI-SOP in a dry state, pre-wetted, and submersed. Additionally, further testing showed the coated fabrics exhibit strong fouling resistance to oils, and maintained their SHI-SOP properties, even after 1000 cycles of Martindale abrasion testing, Figure 5b. The authors attributed the SHI-SOP characteristics of the coated cotton to the increased surface roughness imparted by the SiNPs and the presence of hydrophilic and oleophobic functional groups.

Recently, Jang and co-workers utilized F-SiNPs as an anti-soiling coating for solar mirrors [48]. In this example, the authors coated solar mirrors with a SiNPs suspension using a draw down coater. Once bound to the surface, the SiNPs were functionalized using the thermal vapor deposition of (heptadecafluoro-1,1,2,2,-tetrahudrodecyl)trichlorosilane. The coated mirrors were determined to be superhydrophobic with WCA of 165°, Figure 6a, and SEM analysis showed characteristics of a nanotextured surface, Figure 6b. 

The evaluation of optical uniformity of the specular reflectance indicated no impact on the functionality of the solar mirror as a result of the coating. The soiling rate towards the dust of the F-SiNPs-coated solar mirrors were assessed by using a falling sand apparatus at various inclines. It was determined that the coated solar mirrors had better anti-soiling properties when compared to non-coated mirrors. This was assigned to a reduction in the adhesion forces as a result of the superhydrophobicity and surface roughness. Furthermore, the sand was easily removed with air brushing, allowing the mirrors to maintain the desired specular reflectance. Field testing also demonstrated that the coated solar mirrors outperformed the uncoated mirrors.

Additionally, Lee et al. used F-SiNPs to prepare superamphiphobic (SPO)-superamphiphilic (SPI) patterned substrates for use as in a colorimetric bioassays [49]. The F-SiNPs were prepared in a top-down method by functionalizing SiNPs, that were 10–20 nm in diameter, with 1*H*,1*H*,2*H*,2*H*-perfluorooctyltriethoxysilane. In order to prepare the patterned bioassay, a polydimethylsiloxane substrate was spray-coated with an adhesive and the F-SiNPs. Subsequent treatment of the spray coated material using a mask and oxygen plasma treatment yielded a SPO-SPI patterned film, Figure 7a. The substrate was characterized by SEM and XPS and demonstrated that the film had been successfully fluorinated by the F-SiNPs and patterned with the oxygen plasma treatment. Furthermore, drop analysis with various liquids showed that the SPO region had high contact angles, while the SPI region was nearly zero, Figure 7b. 

Due to the unique surface chemistry and morphology of the stretchable SPO-SPI patterned film, liquid droplets on the surface could easily be manipulated for mixing, transportation, and dispensing. The authors demonstrated that the SPO-SPI substrate could successfully be used to test for biological analytes; including glucose (GL), uric acid (UA), and lactate (LA), Figure 7c. 

Furthermore, Sawada and co-workers prepared F-SiNPs using a bottom-up method [50]. The reactive silanes employed in the co-hydrolysis were *N*-(3-triethoxysilylpropyl)gluconamide and a fluorinated vinyltrimethoxysilane oligomer (R_f_(CH_2_CHSi(OMe)_3_)_n_-R_f_, where R_f_ = CF(CF_3_)OC_3_F_7_ and *M*_n_ = 730), that was synthesized in house. The authors prepared a number of different F-SiNPs where the amount of the fluorinated silane was constant (200 mg) and the amount of *N*-(3-triethoxysilylpropyl)gluconamide was varied (10–1400 mg). Field-emission (FE)-SEM characterization of the particles indicated they were more cubic than spherical, Figure 8b, and dynamic light scattering showed diameters between 32–71 nm.

The authors then used the particles to coat glass substrates and polyethylene terephthalate fabric to investigate the wetting properties of the F-SiNPs using water and dodecane. It was determined that the ability to control the ratio of fluorinated silane to *N*-(3-triethoxysilylpropyl)gluconamide allowed for the ability to tune the wettability of the resulting films from highly oleophobic/superhydrophobic to highly oleophobic/superhydrophilic. 

### 3.2. Membrane Distillation and Separation

The same chemical and physical properties of F-SiNPs, that make them attractive candidates for advanced surface coatings, have also led to their use in membrane distillation (MD) and other membrane separation techniques. These processes have shown a use in the desalinization of water, treatment of waste water, separation of oil and water, and many others. Membrane distillation is a thermal process that is driven by changes in vapor pressure [51]. In order to effectively implement MD, the membrane must exhibit certain properties, such as high thermal stability, low fouling, increased surface roughness and porosity, increased chemical stability, and good permeability [52,53]. Membrane separation is similar to MD, in that it is a purification/separation technique employed for many of the same applications; however, its working principle relies on concentration gradients, changes in absolute pressure, or electrical potential gradients [52,54]. As a result, many of the same properties are still required. 

Recently, Jiang and co-workers prepared a superhydrophobic membrane utilizing F-SiNPs and demonstrated its use in vacuum membrane distillation (VMD) for the treatment of high salinity water [55]. Initially, the authors prepared a SiNPs/polypropylene (PP) composite film by the in-situ sol-gel reaction of TEOS with a PP thin film. The SiNPs were then fluorinated with PFTES in a top-down fashion to obtain the superhydrophobic (F-SiNPs/PP) membrane. An analysis of the films by attenuated total reflection (ATR)-IR and XPS showed successful incorporation of the SiNPs in the films and subsequent fluorination. The surface morphology and surface energy of the F-SiNPs/PP membrane was investigated by SEM, AFM, and WCA analysis. The F-SiNPs/PP membrane was determined to be superhydrophobic with a WCA of 159.0°. The film was evaluated in VMD and found to maintain the desired anti-wetting and anti-fouling properties when a 3.5 wt % sodium chloride (NaCl) feed solution was used, under both continuous and batch operation. At higher concentrations (15 wt %) of NaCl, the film maintained high salt rejection under continuous treatment, with a 20% reduction in the fouling rate. 

In another example, Pang and Zhang prepared a hydrophobic thin film nanocomposite reverse osmosis (TFN RO) membranes, utilizing F-SiNPs [56]. The F-SiNPs were prepared in a bottom-up fashion, with the co-hydrolysis/condensation of TEOS and PFTES. The F-SiNPs were characterized by SEM and found to be 150–200 nm in diameter. The incorporation of fluorine into the particles was confirmed by ATR-IR and FE-SEM. Polyamide TFN RO membranes, containing the F-SiNPs, were prepared using an interfacial polymerization (IP) technique using trimesoyl chloride (TCl) and *m-*phenylenediamine (MDP). The membranes were polymerized onto a polysulfone support and evaluated for desalination performance, Scheme 5. The incorporation of the F-SiNPs in the TFN RO showed a 2.6% increase in the salt rejection when compared to a TFN RO film without the F-SiNPs. 

Additionally, Huang et al. synthesized a fibrous membrane, containing F-SiNPs, for the purpose of gravity driven oil-water separation [57]. In their study, the authors took cellulose-acetate-polyimide (CA-PI) core-sheath nanofibers prepared through electrospinning, and coated them with F-SiNPs by the in-situ polymerization of 2,2-bis(3-*m*-trifluoromethylphenyl-*l*-1,4-dihydro-2*H*-1,3-benzoxazinyl)propane (BAF-tfa) in the presence of SiNPs, Figure 9a. The SiNPs used were between 7–40 nm in diameter. Successful incorporation of the F-SiNPs was verified by FT-IR and by changes in the morphology of the surface by SEM, Figure 9b. Through WCA and oil contact angle (OCA) measurements, it was found that the CA-PI-F-SiNPs nanofibers were superhydrophobic, with WCA as high as 166°C, and superoleophilic, with very low OCA. The WCA and OCA were dependent on concentration of BAF-tfa and the SiNPs used in the functionalization, thus the hydrophobicity and oleophilicity could be tuned, Figure 9c. The ability to use the membrane in the gravity driven separation of oil-water was evaluated using mixtures of dichloromethane/oil/water. The measured flux of the membrane was significantly higher than commercial filtration membranes and showed efficient separation of water from oil. The membranes were also stable over a wide pH range, mechanically robust, thermally stable, and showed increased surface roughness. 

### 3.3. Biological Probes

Clinical ^1^H magnetic resonance imaging (MRI) utilizes the difference in relaxation properties of the water molecules present in biological systems to obtain an image and typically requires contrast agents to improve resolution and sensitivity [58]. An alternative to traditional contrast agents is to employ heteronuclear atoms, like fluorine (^19^F), that exhibit what are known as a “hot spots” in the MRI, providing a more detailed image with improved contrast and resolution. The use of fluorine is also advantageous because of its 100% natural abundance and high MR sensitivity (83% relative to ^1^H). Fluorine also resonates at a similar frequency as ^1^H, so little modification is required to transition from ^1^H to ^19^F detection. Furthermore, because there are only trace amounts of fluorine already in biological tissue, the MRI generated using ^19^F contrast agents will have low background signals [58,59,60]. Silica based nanomaterials for biological applications are also advantageous because they are known to have good biocompatibility, water solubility, low toxicity, a range of pH tolerances, and can be fluorescent [61,62,63]. As a result, F-SiNPs have been utilized for biological probes, including MRI and fluorescent imaging. 

Recently, Zhou and co-workers employed F-SiNPs as an imaging and detection tool for small cell lung tumors in mice [59]. The authors synthesized F-SiNPs using a bottom-up method with (3-aminopropyl)trimethoxysilane and perfluoro-*tert*-butyl alcohol. The particles were prepared by reducing the methoxy groups to silanols, using trisodium citrate dihydrate in a one-pot microwave assisted reaction. The F-SiNPs had a diameter of 3.85 +/− 1.57 nm, and had good solubility in water, with little aggregation as a result of the surface amine groups. The incorporation of fluorine was verified by XPS, energy dispersive X-ray spectroscopy, and FT-IR spectroscopy. In order to use the F-SiNPs as an imaging tool for tumor cells, they were functionalized with the arginylglycylaspartic acid (RGD) peptide, which is known to bind to various types of tumor cells containing integrin α_v_β_3_ (A549), Figure 10a. The functionalized F-SiNPs were then evaluated both in vitro and in vivo. With the former, it was found that the functionalized particles had non- or low-cytotoxicity, and were selectively incorporated into the A549 tumor cells, as indicated by an increase in the fluorescence signal and ^19^F MRI signal from the lysed cells, Figure 10b. In the in vivo studies, the RGD functionalized F-SiNPs were injected into the A549 tumors of mice. The ^1^H MRI showed the anatomic structure and tumor profile, while the ^19^F MRI showed the “hot spot” of the tumor, Figure 10c. Furthermore, histopathologic analysis of the heart, liver, lung, kidney, and spleen indicated no toxic effect on the mice from the F-SiNPs. 

In another recent example, Fortin et al. prepared F-SiNPs using a top-down method from SiNPs that were 140 nm in diameter [60]. Two different fluorinated moieties were used producing two different F-SiNPs, dimethoxy-methyl(3,3,3-trifluoropropyl)silane and poly(methyl-3,3,3-trifluoropropylsiloxane). The particles functionalized with the polyfluorosiloxane also included PEG to aid in dispersion. Each F-SiNPs was further functionalized with diethylenetriaminepentaacetic dianhydride (DTPA), and gadolinium, Figure 11a. Gadolinium is often employed as a traditional MRI contrast agent [58] and known to be chelated by DTPA. The incorporation of the fluorinated compounds, DTPA and PEG, were confirmed by thermogravimetric analysis, XPS, and multinuclear NMR spectroscopy. The analysis supported that the polyfluorosiloxane functionalized particles had more fluorine content. Both types of particles also showed no flocculation or agglomeration while suspended in aqueous solutions and were stable for at least 1 week. With the polyfluorosiloxane F-SiNPs having more fluorine content, it was determined to have a greater enhancing effect on the ^19^F MRI signal. Both types of gadolinium functionalized F-SiNPs showed strong signals in the ^1^H MRI of water, even at low concentrations of gadolinium, Figure 11b. 

## 4. Conclusions and Future Perspective

Fluorinated silica nanometer sized particles are an exceptional class of functional materials. The incorporation of fluorine into a nanometer-sized silica particle yields a hybrid material with inherent low surface energy and enhanced surface roughness. As a result, the use of F-SiNPs, for advanced coatings and membranes, is still quite saturated. However, as the technology has matured, more advanced applications, including those in the biomedical arena, are becoming more prevalent. The vast majority of known synthetic reports utilize the versatile sol-gel technique for obtaining F-SiNPs, with only brief mention of lesser employed methods. This technique is well studied and established, so the ease in which consistent and uniform particles can be obtained is virtually unmatched. Although there are commercially available fluorinated reactive silanes that can be used in the sol-gel method, they are mostly limited to linear per-fluorinated carbon chains, which restricts the chemical variation, and potentially the scope, of the F-SiNPs. Of greater consequence and possibly one reason F-SiNPs have yet to break into the commercial market, are the environmental concerns surrounding long per-fluorinated carbon chains [64,65]. Although this issue is being addressed by some [32,44], there is a growing need for researchers and chemical companies to develop novel reactive silanes that might be used and sold in their place.

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
