# Peer review of "Recent Studies on Fluorinated Silica Nanometer-Sized Particles"

_nanomaterials, 2019, doi:10.3390/nano9050684_

Reviewer 1 Report

This review was aimed to summarize the synthetic method and applications of fluorinated nanometer-sized silica particles. In this review, it was well-described the recent work on F-SiNPs in terms of synthetic methods and applications. I recommend to publish this review in Nanomaterials after minor correction.  

As authors already mentioned that this review was particularly focused on the recent examples in Introduction part, I think that the title “Fluorinated Silica Nanometer-sized Particles: Past, Present, and Future Perspective” is too general and broad. It would be more appropriate if it is “Recent studies on Fluorinated Silica Nanometer-sized Particles” or similar.

And, there are some typing errors (e.g., mico in Page 2) and missing references as marked “Error! Reference source not found” in manuscript (Page 3, 4, 5). Moreover, it should be mentioned the full name of abbreviation at first, then used the abbreviation only after that. “F-SiNPs” and “fluorinated nanometer-sized silica particles” seems to be same words but used both so that it is needed to unify in this manuscript.

Author Response

Comments and Suggestions for Authors

This review was aimed to summarize the synthetic method and applications of fluorinated nanometer-sized silica particles. In this review, it was well-described the recent work on F-SiNPs in terms of synthetic methods and applications. I recommend to publish this review in Nanomaterials after minor correction. 

As authors already mentioned that this review was particularly focused on the recent examples in Introduction part, I think that the title “Fluorinated Silica Nanometer-sized Particles: Past, Present, and Future Perspective” is too general and broad. It would be more appropriate if it is “Recent studies on Fluorinated Silica Nanometer-sized Particles” or similar.

Authors Response to Reviewer: The title was changed to “Recent Studies on Fluorinated Silica Nanometer-sized Particles”

And, there are some typing errors (e.g., mico in Page 2) and missing references as marked “Error! Reference source not found” in manuscript (Page 3, 4, 5). Moreover, it should be mentioned the full name of abbreviation at first, then used the abbreviation only after that. “F-SiNPs” and “fluorinated nanometer-sized silica particles” seems to be same words but used both so that it is needed to unify in this manuscript

Authors Responses to Reviewer: The minor typing error on Page 2, “mico” was changed to “micro”. The document was reviewed for additional typing errors and corrections were made accordingly.

The “Error! Reference source not found” were removed throughout the document.

“Fluorinated nanometer-sized silica particles” was defined in the abstract. This abbreviation was used throughout the document, unless at the beginning of a sentence in the main text. In this case,” Fluorinated nanometer-sized silica particles” was used. Other abbreviations used in the document were double checked to make sure they were defined at first mention. These abbreviations were used, except when appearing at the beginning of a sentence. Any abbreviations appearing only once in the document were removed and only the un-abbreviated text used.

Reviewer 2 Report

This review summarized recent results of fluorine-containing silica and silsesquioxanes. As the authors mentioned, this review does not cover all of the results, but all the more we can easily go through the related fields and obtain necessary information. Therefore, this paper is suitable for publication in Nanomaterialsbecause it meets the standards of this journal in terms of quality 

and significance.;acceptance of the manuscript after minor revision is recommended.

1.   Scheme 2: This scheme shows two-step reaction, and hexafluoro-2-propanol is not the product of the first reaction. Please move this compound besides the arrow of the second reaction.

2.   Page 4, Line 4 from the bottom: RAFT may not be the common word for general readers. Please show the full name here.

3.   Similarly, for SHI-SOP (Page 9, Line 13), explanation appears in the following page, but it may be better to show here.  

4.   Page 8, last paragraph: Could you describe about the initial adhesive strength according to the fluorine content? It can be expected that adhesive strength may be lower for fluorinated compounds because of low surface energy.

Minor points:

>Page 5, Line 6 from the bottom: no space in (3-mercaptopropyl)triethoxysilane. Similar in Page 7, line 3.

>Page 9, Line 4: no space in 1H,1H,2H,2H.

Author Response

Comments and Suggestions for Authors

This review summarized recent results of fluorine-containing silica and silsesquioxanes. As the authors mentioned, this review does not cover all of the results, but all the more we can easily go through the related fields and obtain necessary information. Therefore, this paper is suitable for publication in Nanomaterialsbecause it meets the standards of this journal in terms of quality and significance; acceptance of the manuscript after minor revision is recommended. 

1.     Scheme 2: This scheme shows two-step reaction, and hexafluoro-2-propanol is not the product of the first reaction. Please move this compound besides the arrow of the second reaction.

Response to Reviewer: The scheme was redrawn to include the hexafluoroisopropanol on the second reaction arrow to emphasize its use as a reagent.

2.   Page 4, Line 4 from the bottom: RAFT may not be the common word for general readers. Please show the full name here.

Response to Reviewer: The text was updated to define reversible addition-fragmentation chain transfer (RAFT) at first mention.

3.   Similarly, for SHI-SOP (Page 9, Line 13), explanation appears in the following page, but it may be better to show here.  

Response to Reviewer: superoleophobic-superhydrophilic (SHI-SOP) was defined on Page 9, line 2.

4.   Page 8, last paragraph: Could you describe about the initial adhesive strength according to the fluorine content? It can be expected that adhesive strength may be lower for fluorinated compounds because of low surface

Response to Reviewer: “Although the control had better adhesive strength initially than the F-SiNPs film (1.92 MPa vs. 1.82 MPa), under the humid thermal aging conditions...”, was added to this paragraph.

Minor points:

>Page 5, Line 6 from the bottom: no space in (3-mercaptopropyl)triethoxysilane. Similar in Page 7, line 3.

>Page 9, Line 4: no space in 1H,1H,2H,2H

Response to Reviewer: These and similar corrections were made throughout the document.

Round  2

Reviewer 1 Report

The revised manuscript is suitable for publication in Nanomaterials

Reviewer 2 Report

All the points reviewers commented were carefully considered and revised. 

I think we can accept this review as it is.